# CORRECTION OF DECOUPLED WEIGHT DECAY

## ABSTRACT

Decoupled weight decay, solely responsible for the performance advantage of AdamW over Adam, has long been set to proportional to learning rate $\gamma$ without questioning. Some researchers have recently challenged such assumption and argued that decoupled weight decay should be set $\propto \gamma^2$ instead based on orthogonality arguments at steady state. To the contrary, we find that eliminating the contribution of the perpendicular component of the update to the weight norm leads to little change to the training dynamics. Instead, we derive that decoupled weight decay $\propto \gamma^2$ results in stable weight norm based on the simple assumption that updates become independent of the weights at steady state, regardless of the nature of the optimizer. Based on the same assumption, we derive and empirically verify that the Total Update Contribution (TUC) of a minibatch under the Scion optimizer is better characterized by the momentum-dependent effective learning rate whose optimal value transfers and we show that decoupled weight decay $\propto \gamma^2$ leads to stable weight and gradient norms and allows us to better control the training dynamics and improve the model performance.

## 1 INTRODUCTION

$L_2$ regularization, a common technique for controlling model weight growth and preventing overfitting, is equivalent to weight decay for unmodified SGD. For adaptive gradient methods such as SGD with momentum (Sutskever et al., 2013) and Adam (Kingma & Ba, 2015), weight decay is no longer equivalent to $L_2$ regularization, and empirical observations have led to the development of the decoupled weight decay of AdamW (Loshchilov & Hutter, 2019) that outperforms the original Adam with the following update rules:

$$\boldsymbol{g}_t \leftarrow \nabla_\theta f_t(\boldsymbol{\theta}_{t-1})$$
$$\boldsymbol{m}_t \leftarrow \beta_1 \boldsymbol{m}_{t-1} + (1 - \beta_1)\boldsymbol{g}_t$$
$$\boldsymbol{v}_t \leftarrow \beta_2 \boldsymbol{v}_{t-1} + (1 - \beta_2)\boldsymbol{g}_t^2$$
$$\boldsymbol{u}_t \leftarrow \frac{\boldsymbol{m}_t/(1 - \beta_1^t)}{\sqrt{\boldsymbol{v}_t/(1 - \beta_2^t)}}$$
$$\boldsymbol{\theta}_t \leftarrow \boldsymbol{\theta}_{t-1} - \gamma\left(\lambda\boldsymbol{\theta}_{t-1} + \boldsymbol{u}_t\right)$$

where squaring and division are understood to be element-wise, $\theta_t$ and $f_t$ are the model weights and loss function, $m_t$ and $v_t$ are the first and second moments of the loss gradient $g_t$, $u_t$ is the parameter update, and learning rate $\gamma$, weight decay coefficient $\lambda$, betas $(\beta_1, \beta_2)$ and epsilon $\epsilon$ are the hyperparameters. Accordingly, we get the following expression for the expected value of the $l^2$-norm squared of the layer weight vectors:

$$\mathbb{E}[||\boldsymbol{\theta}_t||^2] = \mathbb{E}[||(1 - \gamma\lambda)\boldsymbol{\theta}_{t-1} - \gamma\boldsymbol{u}_t||^2]$$
$$= \mathbb{E}[(1 - \gamma\lambda)^2||\boldsymbol{\theta}_{t-1}||^2 + \gamma^2||\boldsymbol{u}_t||^2 - 2\gamma(1 - \gamma\lambda)\langle\boldsymbol{\theta}_{t-1}, \boldsymbol{u}_t\rangle] \quad (1)$$

Kosson et al. (2024) argues that the changes of model weights can be modeled as random walk and at steady state. If we assume that as $t \to \infty$, $\mathbb{E}[||\boldsymbol{u}_t||^2]$ becomes a time-independent constant $C$ and $\mathbb{E}[\langle\boldsymbol{\theta}_{t-1}, \boldsymbol{u}_t\rangle] = 0$ since $\boldsymbol{\theta}_{t-1}$ and $\boldsymbol{u}_t$ are independent, then

$$\mathbb{E}[||\boldsymbol{\theta}_t||^2] = \mathbb{E}[(1 - \gamma\lambda)^2||\boldsymbol{\theta}_{t-1}||^2 + \gamma^2 C]$$

At steady state $\mathbb{E}[||\boldsymbol{\theta}_t||^2] = \mathbb{E}[||\boldsymbol{\theta}_{t-1}||^2]$, we can solve for $\mathbb{E}[||\boldsymbol{\theta}_t||^2]$:

$$\mathbb{E}[||\boldsymbol{\theta}_t||^2] = \frac{\gamma C}{\lambda(2 - \gamma\lambda)} \approx \frac{\gamma C}{2\lambda} \quad (2)$$

---

**Algorithm 1** "Renormalized" AdamW

---

1: **Input:** Initial values $\boldsymbol{\theta}_{0,l}$ for all layers $l$,
2: **Input:** scheduled learning rate $\gamma_t$, weight-decay coefficient $\lambda, (\beta_1, \beta_2), \epsilon$
3: $\boldsymbol{v}_{0,l} = \boldsymbol{m}_{0,l} = 0$
4: **for** $t = 1$ **to** $T$ **do**
5:     **for** layer $l = 0$ **to** $L$ **do**
6:         $\boldsymbol{g}_{t,l} = \nabla_{\theta_l} f_t(\boldsymbol{\theta}_{t-1,l}, \boldsymbol{\zeta}_t)$                                         $\triangleright$ Minibatch gradient
7:         $\boldsymbol{m}_{t,l} = \beta_1 \boldsymbol{m}_{t-1,l} + (1 - \beta_1)\boldsymbol{g}_{t,l}$
8:         $\boldsymbol{v}_{t,l} = \beta_2 \boldsymbol{v}_{t-1,l} + (1 - \beta_2)\boldsymbol{g}_{t,l}^2$
9:         $\boldsymbol{u}_{t,l} = \frac{\boldsymbol{m}_{t,l}/(1-\beta_1^t)}{\sqrt{\boldsymbol{v}_{t,l}/(1-\beta_2^t)}}$
10:        $\boldsymbol{\theta}_{t-1,l} = \boldsymbol{\theta}_{t-1,l} - \gamma_t \lambda \boldsymbol{\theta}_{t-1,l}$
11:        $\boldsymbol{\theta}_{t,l} = \boldsymbol{\theta}_{t-1,l} - \gamma_t \boldsymbol{u}_{t,l}$                                    $\triangleright$ Standard Adam update
12:        **if** $\lVert \boldsymbol{\theta}_{t-1,l} \rVert \geq \epsilon$ **then**
13:            $u_{t,l\parallel} = \frac{\langle \boldsymbol{\theta}_{t-1,l}, \boldsymbol{u}_{t,l} \rangle}{\lVert \boldsymbol{\theta}_{t-1,l} \rVert}$
14:            ${\color{red}\boldsymbol{\theta}_{t,l} = \frac{\lvert \lVert \boldsymbol{\theta}_{t-1,l} \rVert - \gamma_t u_{t,l\parallel} \rvert}{\lVert \boldsymbol{\theta}_{t,l} \rVert + \epsilon} \boldsymbol{\theta}_{t,l}}$        $\triangleright$ Only keep the contribution of $u_{t,l\parallel}$ to the norm
15:        **end if**
16:     **end for**
17: **end for**

---

Kosson et al. (2024) largely follows the derivation above but further decomposes the update norm into the scalar projection $u_{t\parallel} = \frac{\langle \boldsymbol{\theta}_{t-1}, \boldsymbol{u}_t \rangle}{\lVert \boldsymbol{\theta}_{t-1} \rVert}$ onto the weights and the corresponding scalar rejection $u_{t\perp} = \sqrt{u_t^2 - u_{t\parallel}^2}$. It then argues that since $\mathbb{E}[u_{t\parallel}] = 0$ due to randomness or scale-invariance resulting from normalization, $u_{t\perp}$ drives balanced rotation across all layers at steady state. Defazio (2025) takes a more prudent approach and limits its theory to layers immediately followed by normalization that guarantees $\langle \boldsymbol{\theta}_{t-1}, \boldsymbol{g}_t \rangle = 0$ but comes to a similar conclusion and proposes AdamC, a variant of AdamW that sets $\lambda_t \propto \gamma_t$, the scheduled time-dependent learning rate, for layers followed by normalization to keep the steady-state weight norm constant. Nevertheless, Defazio (2025) presents experiments on Llama 3 architecture (Grattafiori et al., 2024) in which most layers are not immediately followed by normalization. It states that "we consider every linear layer as normalized, excluding the output layer of the network" for the purpose of applying such corrected weight decay, and AdamC results in more stable weight and gradient norms than the AdamW baseline regardless.

In the following sections, we first present experiments showing that $u_{t\perp}$ makes insignificant contributions to the weight norm for pre-norm transformers like Llama 3. We then further generalize the above derivation to constrained Scion (Pethick et al., 2025) and present numerical simulation results as supporting evidence. Finally, we present our experiments showing that ScionC, with $\lambda_t \propto \gamma_t$ analogous to AdamC, exhibits similarly stable weight and gradient norms and improved model performance.

## 1.1 Perpendicular component of the update makes negligible contribution to the weight norm

Consider the "Renormalized" AdamW optimizer above (Algorithm 1) which eliminates the contribution of $u_{t\perp}$ to the weight norm by renormalizing the weights of the layers $l = 0 \ldots L$ by a factor of $\frac{\lvert \lVert \boldsymbol{\theta}_{t-1,l} \rVert - \gamma_t u_{t,l\parallel} \rvert}{\lVert \boldsymbol{\theta}_{t,l} \rVert + \epsilon}$ after update. If the scalar projection $u_{t\parallel}$ is small or zero and the subsequent balanced rotation (Kosson et al., 2024) or gradient-to-weight ratios (Defazio, 2025) are important to the training dynamics, we expect this change to be significant. We train a variant of ViT-S/16 based on the setup described in Beyer et al. (2022) on the ImageNet-1k dataset (Russakovsky et al., 2015) for 90 epochs and instead observe almost no differences in relevant metrics (Fig. 1). Although we cannot exclude the possibility that the balancing effects of AdamW are important for training other classes of models, this contradicting evidence and the fact that AdamW excels at transformer optimization (Zhang et al., 2024) cast doubt on their importance in general.

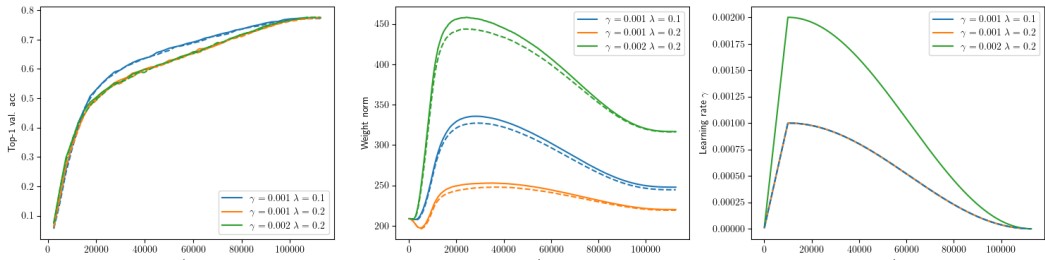

Figure 1: Training a ViT-S/16 with "Renormalized" AdamW results in negligible differences in top-1 val. accuracy (77.15 vs. 77.45 for the $\gamma = 0.001$, $\lambda = 0.1$ AdamW baseline), weight norm, and gradient norm throughout the training process. Notice the suppression of weight norm and surge of gradient norm towards the end of the cosine learning rate decay, characteristic of AdamW. Except using the PyTorch Inception crop with crop scale lower bound $a_{min} = 0.2$, the setup is identical to Beyer et al. (2022).

## 1.2 Expected weight norm with independent weight update at steady state

With evidence against the geometry argument for the steady-state weight norm, let us re-examine the derivation of the steady-state weight norm in Eq. 2. Note that we only assume the existence of a steady state of the weight norm as $t \to \infty$ and that the weight update $\boldsymbol{u}_t$ becomes independent of the model weight $\mathbb{E}[\langle \boldsymbol{\theta}_{t-1}, \boldsymbol{u}_t \rangle] = 0$ at steady state. We make no references to how the optimizer computes the weight update $\boldsymbol{u}_t$ based on the minibatch gradient (Appx. A). We therefore expect the derived steady-state weight norm $\mathbb{E}[||\boldsymbol{\theta}_t||^2] \propto \frac{\gamma C}{2\lambda}$ to be applicable to all optimizers with decoupled weight decay, including SGD with momentum (SGDM) shown in Defazio (2025) and Lion (Chen et al., 2023) discussed in Kosson et al. (2024), as long as they do not violate the stated assumptions. For the remainder of the paper, we further generalize the result to constrained Scion (Pethick et al., 2025) and present Scion with corrected weight decay (ScionC).

## 2 Scion with corrected weight decay

### 2.1 constrained Scion

As formulated in Pethick et al. (2025), the constrained variant of Scion can be considered a collection of optimizers with the following unified update rules. Given layer $l$ and layer weight $\boldsymbol{\theta}_{t,l}$ at time $t-1$, the choice of linear minimization oracle $\text{lmo}_l$, momentum $\alpha$, learning rate $\gamma$, and radius $\rho_l$:

$$\boldsymbol{g}_{t,l} \leftarrow \nabla_{\theta_l} f_t(\boldsymbol{\theta}_{t-1,l}, \zeta_t)$$
$$\boldsymbol{m}_{t,l} \leftarrow (1-\alpha)\boldsymbol{m}_{t-1,l} + \alpha \boldsymbol{g}_{t,l}$$
$$\boldsymbol{\theta}_{t,l} \leftarrow (1-\gamma)\boldsymbol{\theta}_{t-1,l} + \gamma \rho_l \text{lmo}_l(\boldsymbol{m}_{t,l})$$

Table 1 lists the lmos and the norms from which they are derived that we use in our experiments. Conceptually, we choose the norms of the layers based on the shape of the weight and their functions in the model, and lmos are the updates with unit norms in the direction of the steepest descent.

Although equivalent up to reparameterization, the original formulation of Scion deviates significantly from the conventional terminology and makes it difficult to reason about the role of decoupled weight decay in its update rules. We therefore reformulate constrained Scion in terms of independent weight decay coefficient $\eta = \gamma$, layer-wise learning rate $\gamma_l = \gamma \rho_l$, and layer-wise weight decay coefficient $\lambda_l = \frac{1}{\rho_l}$. The update rules then become

$$\boldsymbol{g}_{t,l} \leftarrow \nabla_{\theta_l} f_t(\boldsymbol{\theta}_{t-1,l}, \zeta_t)$$
$$\boldsymbol{m}_{t,l} \leftarrow (1-\alpha)\boldsymbol{m}_{t-1,l} + \alpha \boldsymbol{g}_{t,l}$$
$$\boldsymbol{\theta}_{t,l} \leftarrow (1-\eta)\boldsymbol{\theta}_{t-1,l} + \gamma_l \text{lmo}_l(\boldsymbol{m}_{t,l})$$
$$= \boldsymbol{\theta}_{t-1,l} + \gamma_l \left(-\lambda_l \boldsymbol{\theta}_{t-1,l} + \text{lmo}_l(\boldsymbol{m}_{t,l})\right)$$

Table 1: Norms and the associated lmos as normalized in our experiments. Sign and Spectral assume matrix weight $\boldsymbol{\theta}_l = \boldsymbol{A} \in \mathbb{R}^{d_{\text{out}} \times d_{\text{in}}}$ while Bias assumes vector weight $\boldsymbol{\theta}_l = \boldsymbol{b}_\ell \in \mathbb{R}^{d_{\text{out}}}$. $\boldsymbol{U}\boldsymbol{V}^\top$ refers to the reduced SVD of the input matrix with unitary matrices $\boldsymbol{U}$ and $\boldsymbol{V}^\top$ from the full SVD $\boldsymbol{A} = \boldsymbol{U}\text{diag}(\boldsymbol{\sigma})\boldsymbol{V}^\top$ while $\|\boldsymbol{A}\|_{\mathcal{S}_\infty} = \max(\boldsymbol{\sigma})$ is the spectral norm of the matrix.

|  | Sign | Spectral | Bias |
|---|---|---|---|
| **Norm** | $d_{\text{in}} \max_{i,j} |A_{i,j}|$ | $\sqrt{\frac{d_{\text{in}}}{d_{\text{out}}}}\|\boldsymbol{A}\|_{\mathcal{S}_\infty}$ | RMS |
| **LMO** | $\boldsymbol{A} \mapsto -\frac{\text{sign}(\boldsymbol{A})}{d_{\text{in}}}$ | $\boldsymbol{A} \mapsto -\sqrt{\frac{d_{\text{out}}}{d_{\text{in}}}}\boldsymbol{U}\boldsymbol{V}^\top$ | $\boldsymbol{b}_\ell \mapsto -\frac{\boldsymbol{b}_\ell}{\|\boldsymbol{b}_\ell\|_{\text{RMS}}}$ |

## 2.2 MOMENTUM WITH NORMALIZED UPDATE

So far we have assumed steady-state $\mathbb{E}[\langle \boldsymbol{\theta}_{t-1}, \boldsymbol{u}_t \rangle] = 0$ which implies $\mathbb{E}[\langle \boldsymbol{u}_{t-1}, \boldsymbol{u}_t \rangle] = 0$ for simplicity, even though the use of momentum clearly violates this assumption. Qualitatively, the relationship $\mathbb{E}[\|\boldsymbol{\theta}_t\|^2] \propto \frac{\gamma C}{2\lambda}$ holds regardless since as $\boldsymbol{m}_{t-k,l}$ component of $\boldsymbol{m}_{t,l}$ decays, the update of the far past eventually becomes independent of the current update:

$$\lim_{k \to \infty} \mathbb{E}[\langle \boldsymbol{u}_{t-k}, \boldsymbol{u}_t \rangle] = 0$$

if the minibatch gradients based on which the momentum is updated become independent at the steady state. In the end, we just have a larger constant $C'$ due to the decaying correlation. In fact, if the minibatch gradients $\boldsymbol{g}_t$ become independent with time-independent expected norm at steady state, the second momentum $\boldsymbol{v}_t$ of AdamW stays approximately constant, so the Total Update Contribution (TUC) of the minibatch gradients also remains constant regardless of $\beta_1$ as postulated in Kosson et al. (2024) (Appx. B).

The lmos of Scion normalize the updates so the same reasoning no longer applies and we need to derive $\mathbb{E}[\langle \boldsymbol{\theta}_{t-1}, \boldsymbol{u}_t \rangle]$. Assume that the minibatch gradients become independent with time-independent expected $L_2$ norm $C'$ at steady state, $\mathbb{E}[\langle \boldsymbol{g}_{t'}, \boldsymbol{g}_t \rangle] = C'^2 \delta_{t't}$, where $\delta_{ij}$ is the Kronecker delta function. Then

$$\boldsymbol{m}_t = (1-\alpha)^k \boldsymbol{m}_{t-k} + \alpha \sum_{i=0}^{k-1} (1-\alpha)^i \boldsymbol{g}_{t-i} , \, k \geq 1$$

$$\mathbb{E}[\langle \boldsymbol{m}_{t-k}, \boldsymbol{m}_t \rangle] = C'^2 (1-\alpha)^k$$

In particular $\mathbb{E}[\|\boldsymbol{m}_t\|^2] = C'^2$ so $\mathbb{E}[\|\boldsymbol{m}_t\|_2] = C'$ as expected. Consider the Bias $\text{lmo}_{b_\ell}$ in Table 1 that normalizes the update $\boldsymbol{u}_t = -\text{lmo}_{b_\ell}(\boldsymbol{m}_t) = \frac{\boldsymbol{m}_t}{\|\boldsymbol{m}_t\|_{\text{RMS}}}$. Then

$$\mathbb{E}[\langle \boldsymbol{u}_{t-k}, \boldsymbol{u}_t \rangle] = \mathbb{E}[\frac{\langle \boldsymbol{m}_{t-k}, \boldsymbol{m}_t \rangle}{\|\boldsymbol{m}_{t-k}\|_{\text{RMS}}\|\boldsymbol{m}_t\|_{\text{RMS}}}]$$

Assume that at steady state $\|\boldsymbol{m}_t\| \approx \mathbb{E}[\|\boldsymbol{m}_t\|_2] = C'$. Then

$$\mathbb{E}[\langle \boldsymbol{u}_{t-k}, \boldsymbol{u}_t \rangle] \approx d_{\text{out}}\mathbb{E}[\langle \boldsymbol{m}_{t-k}, \boldsymbol{m}_t \rangle] = d_{\text{out}}(1-\alpha)^k$$

For consistency and brevity, we again denote the $L_2$ norm of the update as $||\boldsymbol{u}_t||_2 = \sqrt{d_{\text{out}}} = C$. We have $\mathbb{E}[\langle \boldsymbol{u}_{t-k}, \boldsymbol{u}_t \rangle] \approx C^2(1-\alpha)^k$. We have

$$\boldsymbol{\theta}_t = (1-\eta)\boldsymbol{\theta}_{t-1} - \gamma \boldsymbol{u}_t$$

$$= -\gamma \sum_{i=0}^{\infty}(1-\eta)^i \boldsymbol{u}_{t-i}$$

$$\boldsymbol{\theta}_{t-1} = -\gamma \sum_{i=0}^{\infty}(1-\eta)^i \boldsymbol{u}_{t-1-i}$$

$$\mathbb{E}[\langle \boldsymbol{\theta}_{t-1}, \boldsymbol{u}_t \rangle] = -\gamma \sum_{i=0}^{\infty}(1-\eta)^i \mathbb{E}[\langle \boldsymbol{u}_{t-1-i}, \boldsymbol{u}_t \rangle]$$

$$= -\gamma C^2 \sum_{i=0}^{\infty}(1-\eta)^i(1-\alpha)^{i+1}$$

$$= -\gamma C^2(1-\alpha) \sum_{i=0}^{\infty}(1-\eta)^i(1-\alpha)^i$$

$$= -\frac{\gamma C^2(1-\alpha)}{1-(1-\eta)(1-\alpha)} = -\frac{\gamma C^2(1-\alpha)}{\eta + \alpha - \alpha\eta}$$

Recall Eq. 1 with the independent weight decay coefficient $\eta = \gamma\lambda$:

$$\mathbb{E}[||\boldsymbol{\theta}_t||^2] = \mathbb{E}[(1-\eta)^2||\boldsymbol{\theta}_{t-1}||^2 + \gamma^2||\boldsymbol{u}_t||^2 - 2\gamma(1-\eta)\langle \boldsymbol{\theta}_{t-1}, \boldsymbol{u}_t \rangle]$$

With $||\boldsymbol{u}_t||^2 = C^2$ and the expression above, at steady state $\mathbb{E}[||\boldsymbol{\theta}_t||^2] = \mathbb{E}[||\boldsymbol{\theta}_{t-1}||^2]$:

$$(2\eta - \eta^2)\mathbb{E}[||\boldsymbol{\theta}_t||^2] = \gamma^2 C^2(1 + 2\frac{(1-\eta)(1-\alpha)}{\eta + \alpha - \alpha\eta})$$

$$\mathbb{E}[||\boldsymbol{\theta}_t||^2] = \frac{\gamma^2 C^2}{2\eta - \eta^2}\frac{(2 - \eta - \alpha + \alpha\eta)}{\eta + \alpha - \alpha\eta}$$

Typically $\eta \ll \alpha \leq 1$. Ignore $O(\eta^2)$ and $O(\eta^3)$ terms of the denominator and $O(\eta)$ terms of the numerator, we get

$$\mathbb{E}[||\boldsymbol{\theta}_t||^2] \approx \gamma^2 C^2 \frac{2-\alpha}{2\alpha\eta} = \frac{\gamma_{\text{eff}}^2 C^2}{2\eta} \tag{3}$$

$$= \gamma C^2 \frac{2-\alpha}{2\alpha\lambda} \tag{4}$$

Eq. 3 again suggests that weight decay should be set $\propto \gamma^2$ and TUC of the minibatch is better characterized by the effective learning rate $\gamma_{\text{eff}} := \gamma\sqrt{\frac{2-\alpha}{\alpha}}$ instead of the raw learning rate $\gamma$ at steady state. Indeed, the optimal effective learning rate $\gamma_{\text{eff}}$ transfers better across different momentum values than the optimal learning rate $\gamma$ (Fig. 2). We can even replace cosine learning rate decay with momentum scheduling for the equivalent $\gamma_{\text{eff}}$ decay throughout most of the training process (Fig. 3, Appx. C). Switching back to the weight decay coefficient $\lambda = \frac{\eta}{\gamma}$, Eq. 4 states that it should be set $\propto \gamma$ for stable weight norm at steady state.

The above derivation applies equally to other $L_2$-norm-based lmos, including ColNorm and RowNorm in Pethick et al. (2025). The Sign $\text{lmo}(\boldsymbol{A}) = -\frac{\text{sign}(\boldsymbol{A})}{d_{\text{in}}}$ is applied element-wise and $-\frac{\text{sign}(A_{i,j})}{d_{\text{in}}} \propto ||A_{i,j}||_\infty = ||A_{i,j}||_2$. It is much more difficult to analyze the dynamics of $\boldsymbol{u}_t$ with the Spectral $\text{lmo}(\boldsymbol{A}) = -\sqrt{\frac{d_{\text{out}}}{d_{\text{in}}}}\boldsymbol{U}\boldsymbol{V}^\top$ but we observe that $\boldsymbol{U}\boldsymbol{V}^\top$ is a semi-orthogonal matrix with Frobenius norm $||\boldsymbol{U}\boldsymbol{V}^\top||_F = \sqrt{\min(d_{\text{in}}, d_{\text{out}})}$. We postulate that the dynamics of $\boldsymbol{u}_t = -\text{lmo}(\boldsymbol{A}) = \sqrt{\frac{d_{\text{out}}}{d_{\text{in}}}}\boldsymbol{U}\boldsymbol{V}^\top$ would be similar to the hypothetical $\boldsymbol{u}'_t = -\text{lmo}'(A) = \sqrt{\frac{d_{\text{out}}}{d_{\text{in}}}\min(d_{\text{in}}, d_{\text{out}})}\frac{\boldsymbol{A}}{||\boldsymbol{A}||_F}$ so Eq. 4 still applies. We therefore propose Scion with corrected weight decay (ScionC, Algorithm 2).

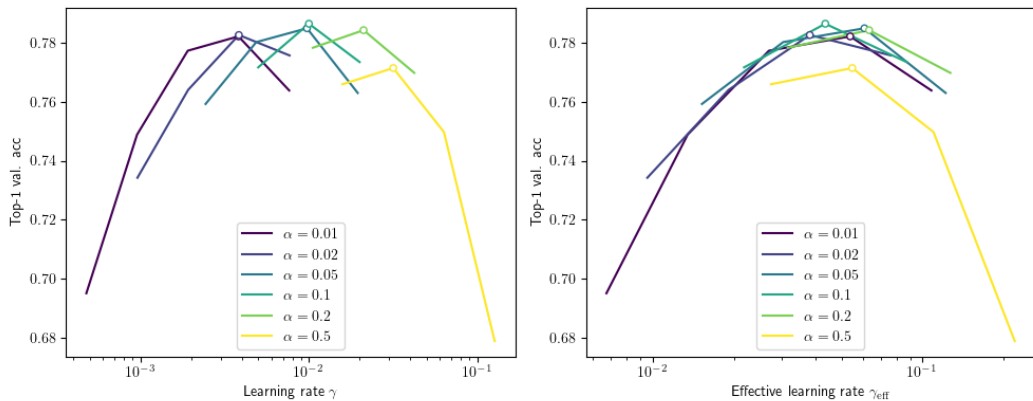

Figure 2: ImageNet-1k top-1 val. accuracy of simple ViT-S/16 trained for 90 epochs with momentum $\alpha \in [0.01, 0.5]$ plotted along the maximum learning rate $\gamma$ (left) vs. maximum steady-state effective learning rate $\gamma_{\text{eff}}$ (right) for the non-Sign parameters at the start of cosine decay. The optimal learning rate $\gamma$ increases with momentum $\alpha$ while the optimal effective momentum $\gamma_{\text{eff}}$ is within a factor of 2 across the momentum values and well within the granularity of the sweep. Weight and gradient norms are kept stable and comparable with ScionC (Algorithm 2 with maximum learning rate $\gamma_L = 0.2$, momentum $\alpha = 0.1$, weight decay coefficient $\lambda_L = 0.004$ for the Sign layer and $C_l^2 = 1.1875$ for other parameters) for these experiments.

---

**Algorithm 2** Scion with corrected weight decay (ScionC)

---

1: **Input:** Initial values $\boldsymbol{\theta}_{0,l}$, layer-wise learning rate schedule $\gamma_{t,l}$, choice of $\text{lmo}_l$ for all layers $l$
2: **Input:** Momentum schedule $\alpha_t$, steady-state norm squared schedule $C_{t,l}^2$ or weight decay coefficient $\lambda_l$ for all layers $l$
3: **for** layer $l = 0$ **to** $L$ **do**
4: $\quad \boldsymbol{m}_{0,l} = 0$
5: **end for**
6: **for** $t = 1$ **to** $T$ **do**
7: $\quad$ **for** layer $l = 0$ **to** $L$ **do**
8: $\quad\quad \boldsymbol{g}_{t,l} = \nabla_{\theta_l} f_t(\boldsymbol{\theta}_{t-1,l}, \boldsymbol{\zeta}_t)$ $\qquad\qquad\qquad\qquad\qquad\qquad$ ▷ Minibatch gradient
9: $\quad\quad \boldsymbol{m}_{t,l} = (1 - \alpha_t)\boldsymbol{m}_{t-1,l} + \alpha_t \boldsymbol{g}_{t,l}$
10: $\quad\quad$ **if** $\lim_{t \to \infty} \mathbb{E}[\langle \boldsymbol{\theta}_{t-1,l}, \boldsymbol{u}_{t,l} \rangle] = 0$ **then**
11: $\quad\quad\quad \lambda_{t,l} = \frac{2 - \alpha_t}{2\alpha_t C_{t,l}^2} \gamma_{t,l}$
12: $\quad\quad$ **else**
13: $\quad\quad\quad \lambda_{t,l} = \lambda_l$
14: $\quad\quad$ **end if**
15: $\quad\quad \boldsymbol{\theta}_{t,l} = \boldsymbol{\theta}_{t-1,l} + \gamma_{t,l} \left( -\lambda_{t,l} \boldsymbol{\theta}_{t-1,l} + \text{lmo}_l(\boldsymbol{m}_{t,l}) \right)$
16: $\quad$ **end for**
17: **end for**

---

## 3 EXPERIMENTS

Our main experiments consist of training a 124M Modded-NanoGPT on FineWeb-Edu-100B (Penedo et al., 2024) with {Scion, ScionC}, PyTorch 2.8 and training the ViT-S/16 described in (Beyer et al. (2022), sometimes called "Simple ViT") on the ImageNet-1k dataset (Russakovsky et al., 2015) with {AdamW, AdamC, Scion, ScionC}, PyTorch 2.5.1 with various training budgets. We use the standard `torch.optim.AdamW` for the AdamW baseline and externally schedule `weight_decay` of the corresponding parameter groups for our AdamC implementation. Our Scion baseline is mostly unmodified from the official implementation of Pethick et al. (2025) except for

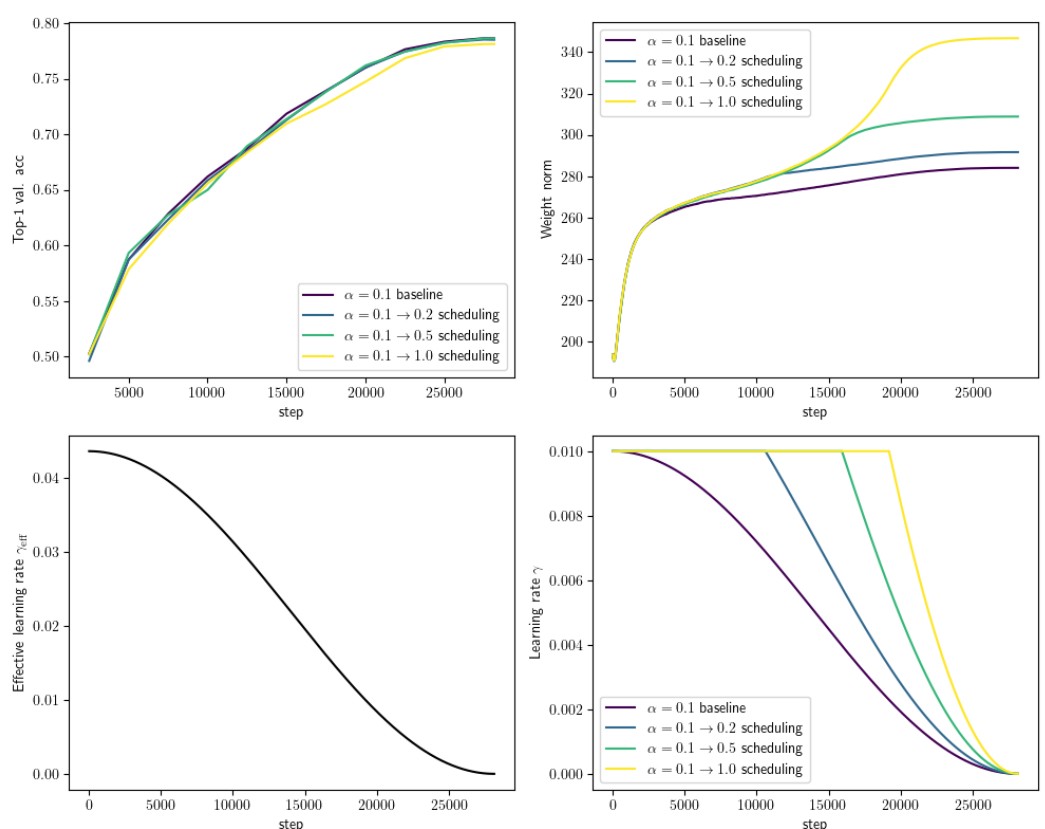

Figure 3: Simple ViT-S/16 trained on ImageNet-1k for 90 epochs with ScionC (Algorithm 2 with maximum learning rate $\gamma_L = 0.2$, momentum $\alpha = 0.1$, weight decay coefficient $\lambda_L = 0.004$ for the Sign layer and maximum learning rate $\gamma = 0.01$, $C_l^2 = 1.1875$ for other parameters) and baseline cosine learning rate decay vs. the equivalent momentum scheduling. For the momentum scheduling experiments $\alpha$ increases from 0.1 to $\alpha_{\max} = \{0.2, 0.5, 1.0\}$ s.t. the effective learning rate $\gamma_{\text{eff}}$ matches that of the cosine learning rate baseline until $\alpha_{\max}$ is reached. The models converge to the same top-1 val. accuracy up till $\alpha_{\max} = 0.5$ where the weight norm approximation starts to break down.

1. The reparameterization described in Sec. 2.1

2. Improvement in efficiency through sharding the state variables and parameter updates on multi-GPU nodes in the spirit of Rajbhandari et al. (2020)

3. Improved reduced SVD accuracy with PolarExpress (Amsel et al., 2025).

We then further modify the multi-GPU Scion to implement ScionC. For the purpose of our experiments, we believe $\lim_{t \to \infty} \mathbb{E}[\langle \boldsymbol{\theta}_{t-1,l}, \boldsymbol{u}_{t,l} \rangle] = 0$ except the output layer (Appx. D). We do not further explore the parameter space of momentum scheduling and instead keep the momentum constant $\alpha = 0.1$ for the main experiments.

## 3.1 MODDED-NANOGPT

For the 124M Modded-NanoGPT experiment, we keep the maximum learning rates from Pethick et al. (2025), $\gamma_L = \gamma\rho_L = 2^{-12} \times 3000$ for the first and last Sign layer (weight-tied), $\gamma_l = \gamma\rho_l = 2^{-12} \times 50$ for the Spectral layers, $\lambda_L = \frac{1}{3000}$ for the Sign layer and $C_l^2 = 5.798$ for the rest for ScionC to keep the initial weight decay the same as the Scion counterpart. We stretch the learning rate schedule with cosine learning rate decay to train the model on the 100B subset of FineWeb-Edu (Penedo et al., 2024). We find that the original batch size $512 \times 1024$ (seqlen) does not fit in the

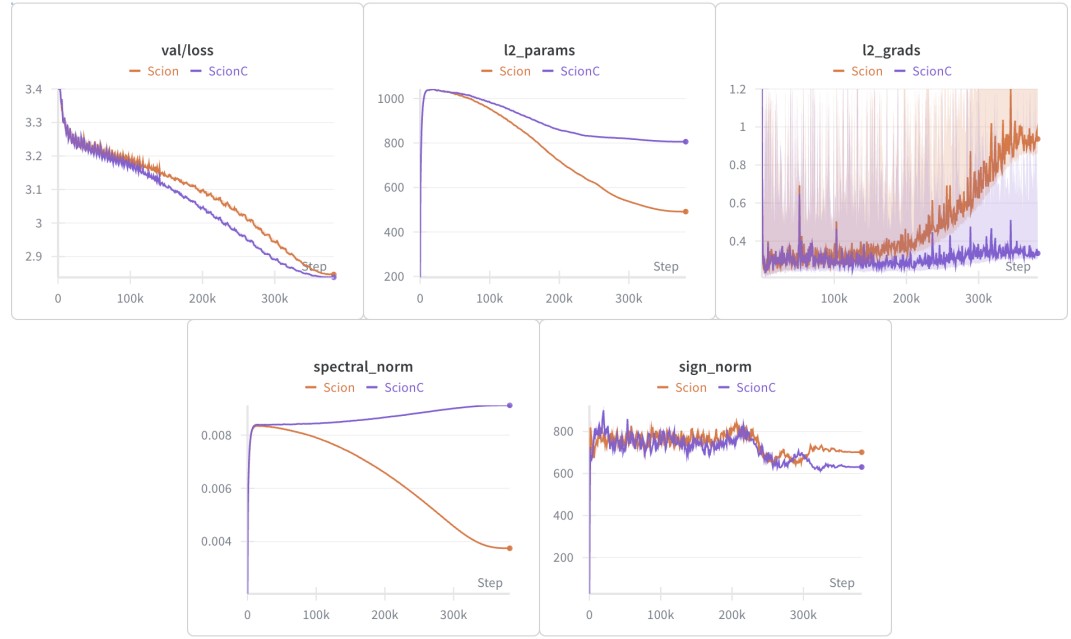

Figure 4: Training 124M Modded-NanoGPT on FineWeb-Edu-100B, Scion vs. ScionC. $\lambda \propto \gamma$ scaling of ScionC results in more stable weight norm, gradient norm, and Spectral norms. The final validation loss is 2.846 for Scion and 2.838 for ScionC.

VRAM of a $8\times$H100 80GB instance and opt to halve the batch size instead of running gradient accumulation. In addition to the typical metrics, we keep track of the Sign norm and the geometric mean of the Spectral norms. We run power iteration (Mises & Pollaczek-Geiringer, 1929) once per step and persist the dominant singular vectors to evaluate the Spectral norms efficiently. We find that ScionC results in lower validation loss (2.838 vs. 2.846) and more stable weight norm, gradient norm, and Spectral norms than the baseline Scion (Fig. 4). The Sign norm is stable in both experiments, in support of the hypothesis that $\lim_{t\to\infty} \mathbb{E}[\langle \boldsymbol{\theta}_{t-1,l}, \boldsymbol{u}_{t,l}\rangle] \neq 0$ for the output layer.

## 3.2 SIMPLE VIT-S/16

For training ViT-S/16 on the ImageNet-1k dataset, we use the model architecture and setup of Beyer et al. (2022) for the {AdamW, AdamC} experiments including sincos2d positional encoding, batch size 1024, global average pooling (GAP), and augmentations including RandAugment (Cubuk et al., 2020) and Mixup (Zhang et al., 2018). The only exception is Inception crop (Szegedy et al., 2015), for which we use the PyTorch implementation with crop scale lower bound $a_{min} = 0.05$. for {AdamW, AdamC, Scion, ScionC}, we train a model for {30, 60, 90, 150, 300} epochs. In addition, we follow the architecture changes made by Pethick et al. (2025) for DeiT (Touvron et al., 2020):

1. Scale the GELU activation function as $\sqrt{2}$GELU to preserve variance
2. Replace LayerNorm with RMSNorm.

We also keep its maximum learning rates $\gamma_L = \gamma\rho_L = 0.0004 \times 500 = 0.2$ for the last Sign layer and $\gamma_l = \gamma\rho_l = 0.0004 \times 25 = 0.01$ for the rest. Overall we find that corrected weight decay requires higher maximum weight decay than the uncorrected counterpart after testing $\lambda \in \{0.1, 0.2\}$ for {AdamW, AdamC} and fully sweeping $\lambda \in \{4 \times 10^{-4}, \mathbf{8 \times 10^{-4}}, 1.2 \times 10^{-3}, 1.6 \times 10^{-3}\}$ for Scion and $C_l^2 \in \{1.1875, \mathbf{0.79167}, 0.59375, 0.475\}$ ($\lambda_L = 0.004$ for the Sign layer) for ScionC (constant). For each setting we repeat the experiment for $N = 3$ random seeds and report the ImageNet-1k top-1 val. accuracy as (mean) $\pm$ (sample standard deviation).

We find this setup of shorter durations in terms of training dynamics than the Modded-NanoGPT experiment. In fact, the model trained with AdamC does not seem to be in steady state even after

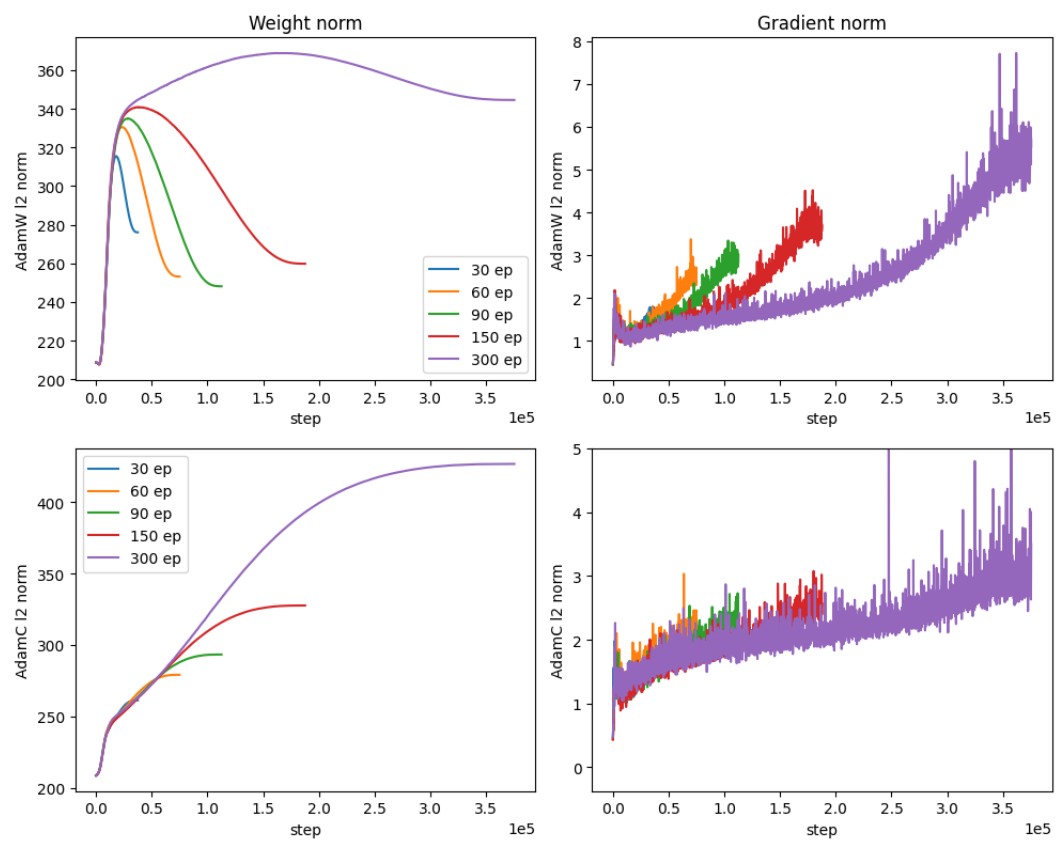

Figure 5: Training ViT-S/16 on ImageNet-1k, AdamW (upper) vs. AdamC (lower). $\lambda \propto \gamma$ scaling of AdamC results in more stable weight and gradient norms. Note that the model does not seem to be in steady state even after 300 epochs.

| | AdamW | AdamC | Scion | ScionC (constant) | ScionC (cosine) |
|---|---|---|---|---|---|
| 30ep | 67.35±0.33 | 67.53±0.27 | **73.31**±0.09 | 73.10±0.18 | 73.10±0.15 |
| 60ep | 74.77±0.08 | 74.59±0.18 | **77.44**±0.09 | 77.20±0.08 | 77.43±0.11 |
| 90ep | 76.92±0.13 | 76.98±0.10 | 78.68±0.09 | 78.53±0.10 | **78.74**±0.09 |
| 150ep | 78.64±0.18 | 78.69±0.03 | **79.65**±0.07 | 79.58±0.04 | 79.62±0.12 |
| 300ep | 79.73±0.12 | 79.70±0.08 | **80.10**±0.14 | 79.94±0.08 | 80.06±0.03 |

Table 2: ImageNet-1k top-1 val. accuracy (original label) of simple ViT-S/16 trained with {AdamW, AdamC, Scion, ScionC} and various training budgets. ScionC models perform as well as the Scion counterparts with more stable weight and gradient norms.

300 epochs (Fig. 5). In contrast, the model trained with ScionC reaches steady state where the model is more likely to benefit (Table 2). Interestingly, Scion holds a slight edge over ScionC (constant), a result that drives us to start scheduling steady-state norm squared $C_{t,l}^2$ to discern at which stage and to what extent it is beneficial to induce weight norm decrease. We test cosine decay of $C_{t,l}^2$ from $C_{0,l}^2 = 1.1875$ to $C_{T,l}^2 = \{\frac{C_{0,l}^2}{2}, \frac{\mathbf{C_{0,l}^2}}{\mathbf{4}}, \frac{C_{0,l}^2}{8}\}$ for ScionC (cosine). ScionC (cosine) matches the performance of Scion, suggesting that the model's performance is indifferent to the detailed schedule of weight norm decrease and the model does not benefit from the terminal weight norm suppression of uncorrected weight decay as $\gamma \to 0$ (Fig. 6), a result that may explain the design choice of non-zero terminal learning rate seen in some literature.

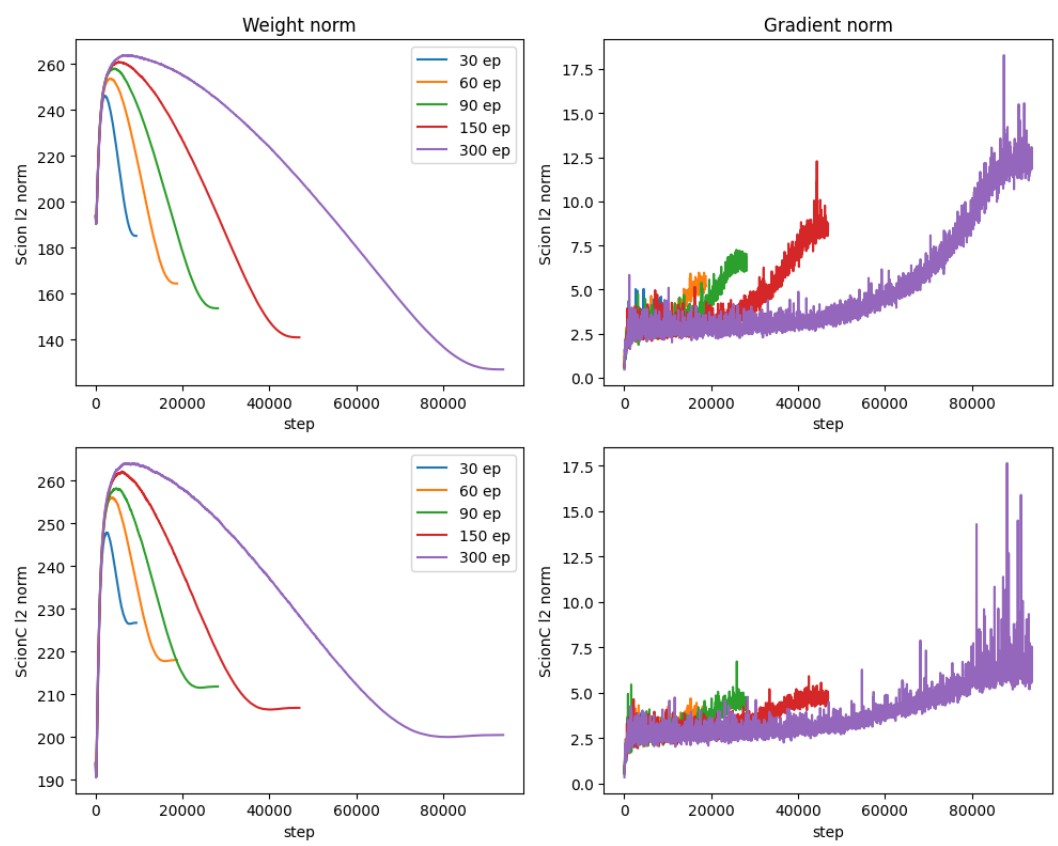

Figure 6: Training ViT-S/16 on ImageNet-1k, Scion (upper) vs. ScionC (cosine, lower). $\lambda \propto \gamma$ scaling of ScionC results in more stable weight and gradient norms.

## 4 RELATED WORK AND CONCLUSION

Due to its importance, the role and effect of weight decay have received much scrutiny (Zhang et al., 2019; D'Angelo et al., 2024; Sun et al., 2025; Kobayashi et al., 2024; Galanti et al., 2025) along with its interactions with the learning rate (Schaipp, 2023) and the sizes of the model and the dataset (Wang & Aitchison, 2025). Paradoxically, its most direct effects on the weight and gradient norms seem to have received less attention (Defazio, 2025; Xie et al., 2023). Furthermore, most of the focus has been on SGD and Adam variants. The Muon optimizer (Jordan et al., 2024b) that can be considered the Spectral-norm subset of unconstrained Scion was in fact proposed without weight decay, likely due to its root in NanoGPT speedrunning (Jordan et al., 2024a). Our result of the dependence of weight decay's effect on momentum (Sec. 2.2) for optimizers with momentum and normalized updates can be considered a major step in resolving their interactions, and we hope that the general random walk model of weight update and decay (Eq. 2) can be further extended to elucidate its role in weight and gradient evolution and model optimization.

### LLM DISCLOSURE

We brainstormed the derivation and approximation of the steady-state weight norm in the case of momentum with normalized update (Sec. 2.2) with DeepSeek R1 (Guo et al., 2025).

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

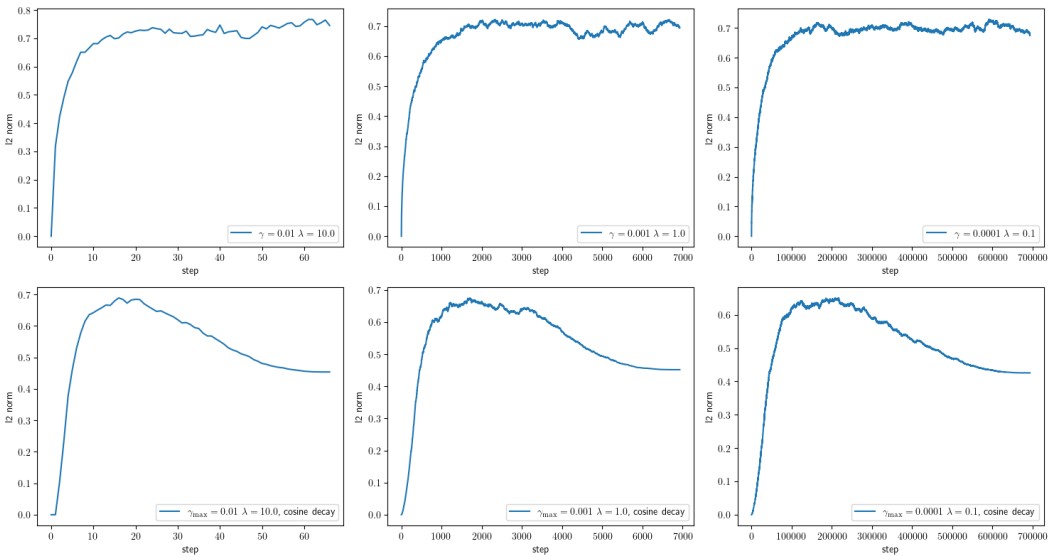

Figure 7: Numerical simulations of the system described by Eq. 5 where $\theta$ is a vector of length $10^3$. $\mathbb{E}[\theta_t^2] = \frac{\gamma C}{2\lambda} = \frac{1}{2000}$ for each element, so the expected $L_2$ norm of the vector is $\approx 0.71$ if we keep the learning rate constant (upper) as expected. If we apply cosine learning rate decay (lower), weight norm decreases towards the end. Here we consistently simulate the system for 10 half-lives $t_{1/2} = -\frac{\log 2}{\log(1-\gamma\lambda)}$, with $0.5\,t_{1/2}$ of linear-warmup and $9.5\,t_{1/2}$ of cosine learning rate decay, so the behavior of the systems looks identical despite $4$ orders of magnitudes of difference in scale.

Hongyi Zhang, Moustapha Cisse, Yann N. Dauphin, and David Lopez-Paz. mixup: Beyond empirical risk minimization. In *International Conference on Learning Representations*, 2018. URL https://openreview.net/forum?id=r1Ddp1-Rb.

Yushun Zhang, Congliang Chen, Tian Ding, Ziniu Li, Ruoyu Sun, and Zhi-Quan Luo. Why transformers need adam: A hessian perspective. In *The Thirty-eighth Annual Conference on Neural Information Processing Systems*, 2024. URL https://openreview.net/forum?id=X6rqEpbnj3.

## A   NUMERICAL SIMULATIONS

Consider the following system where $\theta$ is initialized as $\theta_0 = 0$:

$$\boldsymbol{\theta}_t \leftarrow \boldsymbol{\theta}_{t-1} - \gamma\left(\lambda\boldsymbol{\theta}_{t-1} + \mathcal{N}(0,1)\right) \tag{5}$$

It turns out that this simple system is sufficient to replicate the weight norm behavior towards the end of the cosine learning rate decay, suggesting that the nature of the optimizer is not fundamental to such phenomena (Fig. 7).

## B   BETAS' EFFECT ON THE WEIGHT DECAY AND STEADY-STATE NORM FOR ADAMC

We train a ViT-S/16 on the ImageNet-1k dataset (Russakovsky et al., 2015) for 90 epochs with AdamC and $\beta_1 = \beta_2 = 0.99$ instead of $(\beta_1, \beta_2) = (0.9, 0.999)$ of the main experiment, partially motivated by Orvieto & Gower (2025) (Fig. 8). As predicted, changing betas has no effect on the weight decay and steady-state norm.

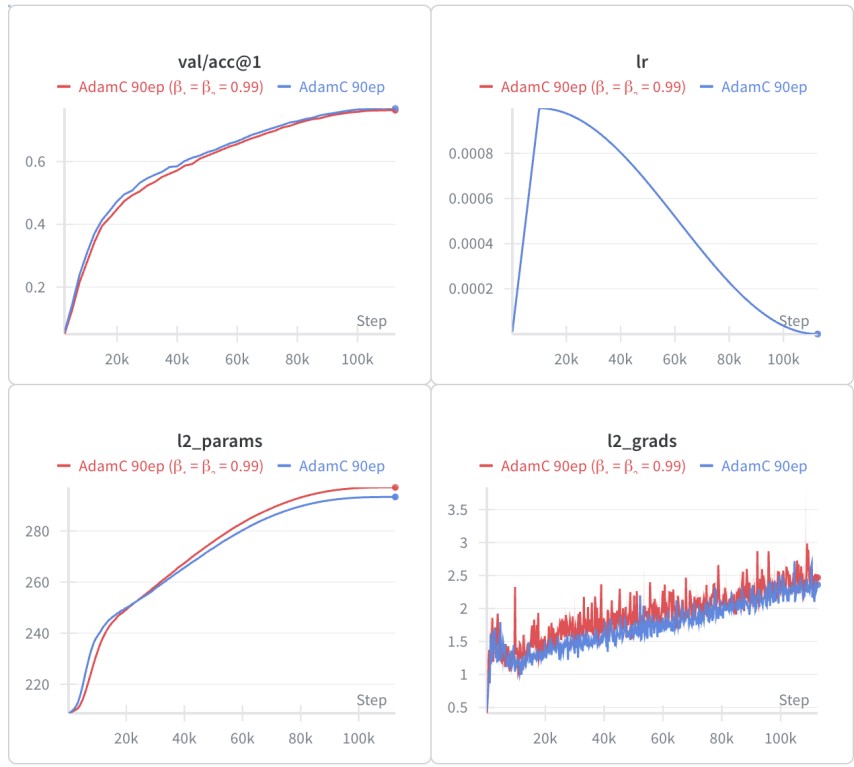

Figure 8: Training a ViT-S/16 on ImageNet-1k for 90 epochs, AdamC with $\beta_1 = \beta_2 = 0.99$ vs. AdamC with $(\beta_1, \beta_2) = (0.9, 0.999)$. Changing the beta values has almost no effects on the weight norm.

## C  ADDITIONAL SCIONC MOMENTUM SCHEDULING EXPERIMENTS

We have run more exploratory experiments to verify Eqs. 3 & 4 by training a ViT-S/16 on the ImageNet-1k dataset (Russakovsky et al., 2015) for 90 epochs with momentum scheduling. Most of the experiments can be explained by comparing their effective learning rate schedule to the cosine learning rate decay baseline.

### C.1  SMALL DEVIATION FROM COSINE LEARNING RATE DECAY

These experiments train the same Simple ViT-S/16 on ImageNet-1k for 90 epochs with ScionC (Algorithm 2) and the same hyperparameters (maximum learning rate $\gamma_L = 0.2$, momentum $\alpha = 0.1$, weight decay coefficient $\lambda_L = 0.004$ for the Sign layer and maximum learning rate $\gamma = 0.01$, $C_l^2 = 1.1875$ for other parameters) as the ones in Fig. 3 but we match $\gamma'_{\text{eff}} = \gamma \frac{2-\alpha}{\alpha}$ of the cosine learning rate baseline with momentum scheduling instead (Fig. 9). Clearly $\gamma'_{\text{eff}}$ is not the correct effective learning rate and it is apparent that the resulting small deviation from the cosine schedule affects the top-1 val. accuracy curves when we consider the correct $\gamma_{\text{eff}}$ of these experiments.

### C.2  MOMENTUM 0.02, SMALL DEVIATION FROM COSINE LEARNING RATE DECAY

These experiments are run with the same setup as those in the previous section but with starting momentum $\alpha = 0.02$ (Fig. 10). Since we erroneously match $\alpha = 0.1, \gamma = 0.01, \gamma'_{\text{eff}} = \gamma \frac{2-\alpha}{\alpha}$ with $\alpha = 0.02$, the correct effective learning rate is too low and the models underperform.

### C.3  LINEAR MOMENTUM SCHEDULING

For this set of experiments, we compare training the same Simple ViT-S/16 on ImageNet-1k for 90 epochs with the following:

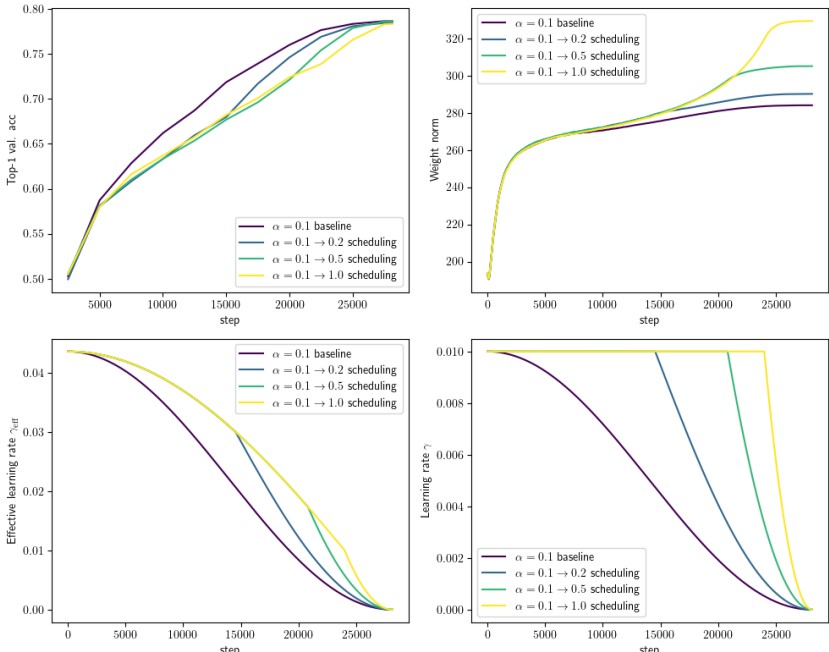

Figure 9: Training a ViT-S/16 with momentum scheduling that erroneously matches $\gamma'_{\text{eff}} = \gamma\frac{2-\alpha}{\alpha}$ of the cosine learning rate baseline. Delayed decay of the correct effective learning rate $\gamma_{\text{eff}} = \gamma\sqrt{\frac{2-\alpha}{\alpha}}$ results in lower top-1 val. accuracy until the very end.

1. The baseline ScionC with $\gamma = 0.01, \alpha = 0.1$, $\eta = 4 \times 10^{-4}$, therefore $\lambda = 0.04$ and $C_l^2 = 2.375$.

2. The $\alpha = 0.01 \to 1.0$ ScionC linear scheduling experiment that linearly increases the momentum in addition to cosine learning rate decay with the same maximum learning rate $\gamma = 0.01$.

3. The $\alpha = 0.01 \to 1.0$ linear scheduling experiment that linearly increases the momentum in addition to cosine learning rate decay but only scales $\lambda \propto \gamma$, ignoring the momentum schedule.

The results are mostly expected if we consider the effective learning rate $\gamma_{\text{eff}}$ over time (Fig. 11). $\gamma_{\text{eff}}$ decays early at the beginning of the $\alpha = 0.01 \to 1.0$ ScionC experiment, so the top-1 val. accuracy rises early at the beginning but soon plateaus while the weight and gradient norms are kept stable with ScionC. $\gamma$ scheduling alone is insufficient to keep weight and gradient norms stable, so they end up swinging drastically for Experiment 3. Interestingly, it eventually converges to higher accuracy, possibly due to its lower weight norm compensating for the vanishing $\gamma_{\text{eff}}$.

## D  OUTPUT LAYER STEADY STATE

In agreement with Defazio (2025), we also come to the conclusion that the learning rate scaling $\lambda \propto \gamma$ should not be applied to the output layer if we are training the model with cross-entropy loss. However, we believe that the reason is not the lack of a subsequent normalization layer but that $\mathbb{E}[\langle \theta_{t-1}, u_t \rangle] \neq 0$ at steady state for the output layer. Say, we have $v = Ax + b$ as the output logits and the model makes the correct prediction for this sample

$$\underset{i}{\operatorname{argmax}}\, v_i = c$$

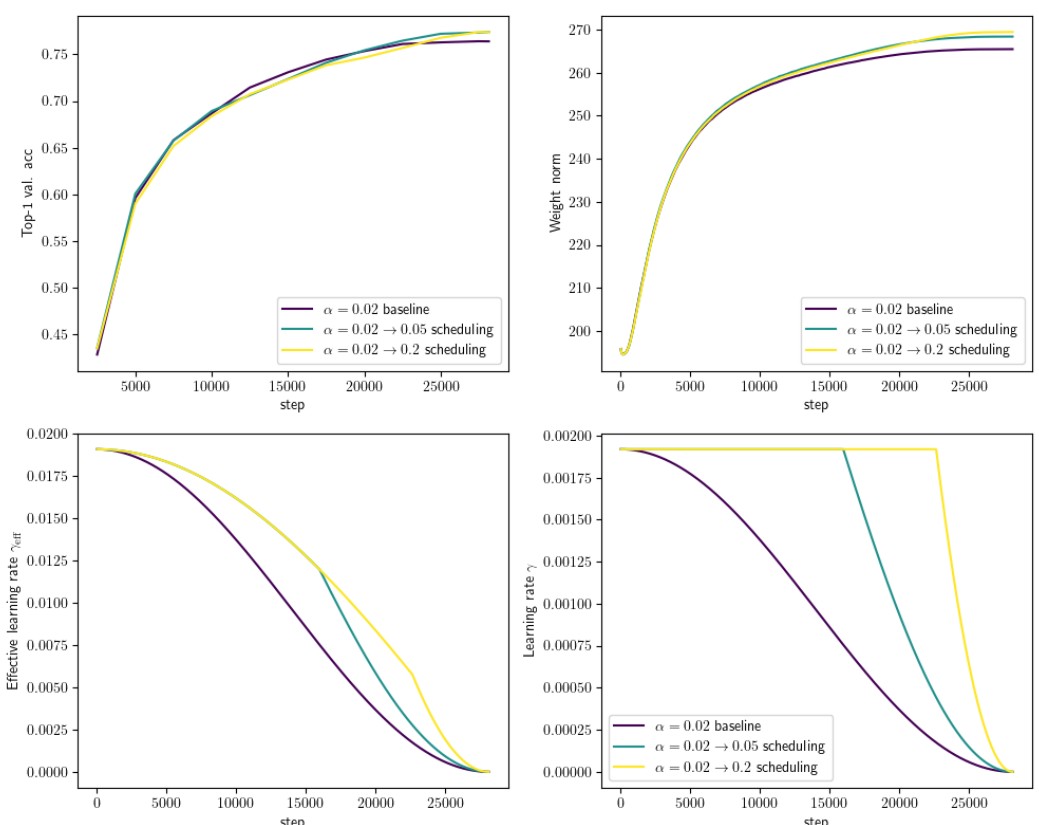

Figure 10: Training a ViT-S/16 with momentum scheduling that erroneously matches $\gamma'_{\text{eff}} = \gamma \frac{2-\alpha}{\alpha}$ of the cosine learning rate baseline and starting momentum $\alpha = 0.02$. The correct effective learning rate is too low for these experiments and delayed decay ends up beneficial.

Then the cross-entropy loss becomes

$$L_{CE,v} = -\log\left(\frac{e^{v_c}}{\sum_i e^{v_i}}\right) = -\log\left(\frac{1}{\sum_i e^{(v_i - v_c)}}\right)$$

Since $\operatorname*{argmax}_i v_i = c$, $\forall_{i \neq c}(v_i - v_c) < 0$. So if we increase $v$ by a small fraction $v' = (1 + \epsilon)v$, $0 < \epsilon \ll 1$:

$$L_{CE,v'} = -\log\left(\frac{1}{\sum_i e^{(v'_i - v'_c)}}\right) = -\log\left(\frac{1}{\sum_i e^{(v_i - v_c)}e^{\epsilon(v_i - v_c)}}\right) < L_{CE,v}$$

By linearity, $v' = A'x + b'$ where $A' = (1+\epsilon)A$, $b' = (1+\epsilon)b$. So, as the model makes more and more correct predictions, the steepest descent increasingly aligns with the weights.[1] $\mathbb{E}[\langle \theta_{t-1}, u_t \rangle]$ is likely to continue to increase, especially if $u_t$ is normalized (Fig. 12).

---

[1] This reasoning suggests that we should also remove the $\lambda \propto \gamma$ dependence of the weight decay of the output layer bias even though we did not for our experiments. We do not expect the difference to be significant.

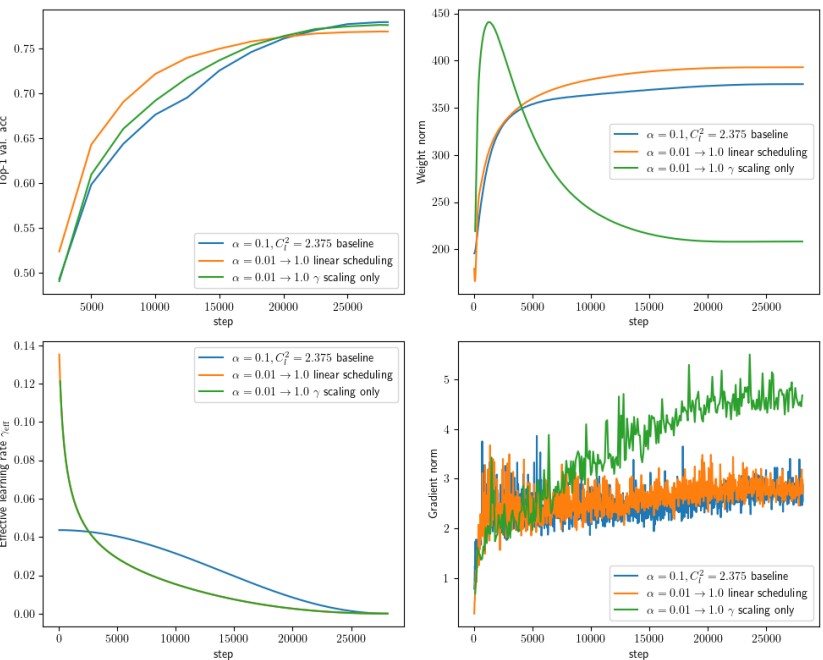

Figure 11: Stress testing ScionC by training a ViT-S/16 with momentum scheduling. Properly scaled and adaptive weight decay results in stable weight and gradient norms, while the learning rate scaling $\lambda \propto \gamma$ alone turns out to be insufficient.

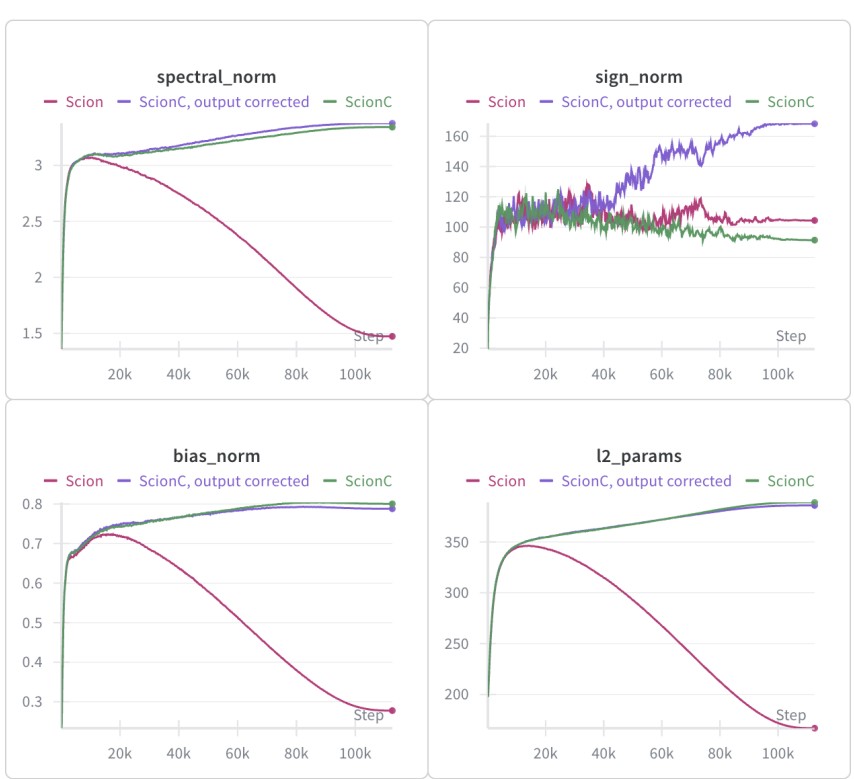

Figure 12: Comparing Scion, ScionC, and ScionC that scales $\lambda \propto \gamma$ for model weights including the output layer while training a ViT-S/16 on the ImageNet-1k dataset (Russakovsky et al., 2015) for 90 epochs. In addition of $L_2$ norm of the model weight, we keep track of the geometric mean of the Spectral norms, arithmetic mean of the Bias norms, and the Sign norm as defined in Table 1 for these experiments. The behavior of the Sign norm is qualitatively different from the others: It continues to increase towards the end of the cosine learning rate decay if we apply the $\lambda \propto \gamma$ correction but remains stable if not corrected.

