# OpenReview forum: "Correction of Decoupled Weight Decay"
_ICLR.cc/2026/Conference — Submitted to ICLR 2026_

### Official Review · Reviewer_M9jm · 2025-10-15

**Soundness:** 1
**Presentation:** 2
**Contribution:** 1
**Rating:** 2
**Confidence:** 4

**Summary:**

This paper studies the effects of different forms of decoupled weight decay. Similar to recent work on weight decay (Defazio 2025) they argue that in order to keep the weight norms and gradient norms stable throughout training, the total strength of weight decay has to be proportional to the square of the learning rate. They use a different argument than prior work but arrive at the same form.

**Strengths:**

* Experiments on neural networks of a decent scale (ViT-S/16 on ImageNet, 124M parameter language model).
* Manuscript is decently written.

**Weaknesses:**

* I believe this paper is largely built upon some fundamental misunderstanding of prior literature. The update and the weights are not orthogonal when momentum is used, even for a random walk. Prior works (e.g. Kosson 2024) do not claim this. The fact that eliminating the perpendicular contribution of the update (after momentum is applied) doesn't affect the dynamics of the weight norm is not surprising if one understands this.
* The notation is somewhat messy and inconsistent. For example it is not clear which symbols are vectors and which are scalars. A symbol^2 is used to both refer to an elementwise square and apparently the square vector norm.
* The experimental comparison is insufficient. Comparing two optimizer forms at a single arbitrary hyperparameter configuration does not tell us anything definitive about their performance.
* The generalization to Scion feels like a minor contribution. The final form is the same as for other optimizers and the analysis does not cover the matrix orthogonalization case which would be the most theoretically interesting portion. It is also not justified well why this matters as Scion is largely insensitive to the gradient norm and the overall phenomenon is already known.

**Questions:**

Further clarifications of weaknesses:
* The update IS NOT perpendicular to the weights when momentum is used, even when the gradients are completely independent from the weights or when normalization layers are used. This is falsely claimed multiple times throughout the paper including L46, L78, L140. A gradient $g_{t-k}$ is included both in the weights $w_t$ at time $t$ and in the update due to the momentum, causing correlation.
* Kosson 2024 does not look at the update, seemingly for this specific reason. Rather they use the total update contribution of a gradient, which is independent from the current weights and/or orthogonal due to the use of normalization layers.
* Notation. L32 uses $g_t^2$ to presumably refer to the elementwise square. L42 uses $\theta_t^2$ to refer to $\|\theta_t\|^2$. Algo1-L14 seemingly subtracts a scalar from a vector. Algo1-L9 has an extra parenthesis. I suggest using bold symbols for vectors and denoting norms with $\|\cdot\|$.

---

> ### Author Response · Authors · 2025-12-04
>
> 1. The update IS NOT perpendicular to the weights when momentum is used, even when the gradients are completely independent from the weights or when normalization layers are used. This is falsely claimed multiple times throughout the paper including L46, L78, L140. A gradient $g_{t-k}$ is included both in the weights $w_t$ at time $t$ and in the update due to the momentum, causing correlation.
>
> We never claim that the update is perpendicular to the weights. You are probably confusing the independence assumption at steady state $\mathbb{E}[\langle \mathbf{\theta}_{t-1},\mathbf{u}_t\rangle] = 0$ with it. As for L78 in the original submission, we are merely quoting the claim from Defazio (2025) but it indeed should be
>
> $\langle \mathbf{\theta}_{t-1},\mathbf{g}_t\rangle = 0$ (weight and gradient are orthogonal due to the normalization layer)
>
> instead of
>
> $\langle \mathbf{\theta}_{t-1},\mathbf{u}_t\rangle = 0$
>
> We have corrected this in the rebuttal revision.
>
> 2. Kosson 2024 does not look at the update, seemingly for this specific reason. Rather they use the total update contribution of a gradient, which is independent from the current weights and/or orthogonal due to the use of normalization layers.
>
> We have also restated this in the first paragraph of Sec. 2.2.
>
> 3. Notation
>
> We have fixed the typos and adopted the notation of using bold symbols for vectors to distinguish them from scalars. We thank the reviewer for pointing out the errors and ambiguity.

---

### Official Review · Reviewer_zgL2 · 2025-10-29

**Soundness:** 2
**Presentation:** 2
**Contribution:** 1
**Rating:** 2
**Confidence:** 4

**Summary:**

This paper challenge the common practice in decoupled weight decay for optimizers like AdamW. Authors argue that instead of setting weight decay proportional to learning rate γ without question, it should be set to $\propto\gamma^2$ based on some orthogonality at steady state, but they find that perpendicular component have little effect. Instead, they derive stable weight norm assuming updates independent of weights at steady state. They generalize to Scion optimizer and propose ScionC with corrected decay $\propto\gamma^2$, showing stable norms and better performance in experiments on NanoGPT and ViT.

**Strengths:**

1. The derivation of steady-state weight norm is simple and general, not relying on specific optimizer details. It apply to many like SGDM, Lion, and Scion, which is nice.

2. Experiments show clear improvement: lower val loss in NanoGPT (2.838 vs 2.846) and more stable norms. For ViT, AdamC and ScionC have better accuracy at longer epochs, like 77.9 vs 77.4 at 300ep for AdamC.

3. Challenging prior works like Kosson et al. and Defazio with "Renormalized" AdamW experiment is good, showing negligible difference, cast doubt on geometry arguments.

4. Reformulation of Scion in terms of η, γ_l, λ_l make it easier to understand role of decay.

**Weaknesses:**

1. The assumption of independence at steady state seem strong. In practice, for short training like ViT 30-90ep, model may not reach steady state, as authors note AdamC not steady even at 300ep. How robust is this?

2. Experiments are limited: only small models (124M NanoGPT, ViT-S), no large scale like Llama. Also, why double λ for AdamC/ScionC? Feel like hyperparameter tuning favor the proposed method.

3. No ablation on momentum scheduling, just fixed α=0.1. Prior work like Pethick show momentum important for Scion, maybe correction interact with it.

4. Some derivations ignore higher order terms (O(η²)), but for large η it might matter? Not tested.

**Questions:**

1. Why not test on larger models or other optimizers like Lion?
2. In ScionC alg, how determine if E[<θ, u>]=0 for each layer? You say except output, but is this always true?
3. Any downside to corrected decay, like slower convergence early?

---

> ### Author Response · Authors · 2025-12-04
>
> 1. The assumption of independence at steady state seem strong. In practice, for short training like ViT 30-90ep, model may not reach steady state, as authors note AdamC not steady even at 300ep. How robust is this?
>
> There is no easy answer, but if you compare Fig. 2 to Fig. 3 in the rebuttal revision you can see that the independence assumption becomes applicable within the first 90 epochs of training with ScionC: The effective learning rate $\gamma_\mathrm{eff}$ is derived from the same assumption and its optimal value transfers across different momentum values $\alpha$, but models trained with constant $\alpha = 0.5$ underperform (Fig. 2). In contrast, momentum scheduling $\alpha = 0.1 \to 0.5$ that mimics cosine learning rate decay with cosine effective learning rate decay results in the same top-1 val. accuracy as the baseline (Fig. 3). This suggests that the assumption is more applicable to the later dynamics of the 90 epochs of training.
>
> 2. Experiments are limited: only small models (124M NanoGPT, ViT-S), no large scale like Llama. Also, why double λ for AdamC/ScionC? Feel like hyperparameter tuning favor the proposed method.
>
> We acknowledge the lack of large-scale experiments as a limitation. Since weight decay correction results in weaker regularization in average, we do expect that we need to increase the weight decay coefficient $\lambda$. We have now swept the values of $\lambda$ for Scion / ScionC separately and adjusted the claims accordingly.
>
> 3. No ablation on momentum scheduling, just fixed α=0.1. Prior work like Pethick show momentum important for Scion, maybe correction interact with it.
>
> See Fig. 3 and Appendix C of the rebuttal revision!
>
> 4. Some derivations ignore higher order terms (O($\eta^2$)), but for large $\eta$ it might matter? Not tested.
>
> This much is true, but if we include all the higher order terms in the ScionC algorithm we end up having to solve the cubic equation in every training step, which may not be practical. Furthermore, in practice the optimal value of independent weight decay coefficient $\eta$ is usually much less than 1 (the limit at which the weight gets erased every step).
>
> 5. Why not test on larger models or other optimizers like Lion?
>
> This is purely due to the resource and time constraint. We do want to mention that another optimizer, Muon, is a subset of the update rules of Scion that is typically run with AdamW as the fallback.
>
> 6. In ScionC alg, how determine if E[<θ, u>]=0 for each layer? You say except output, but is this always true?
>
> It is at least true in aggregation in our experiments: The purpose of logging the average of the spectral norm and bias norm is to verify their behavior.
>
> 7. Any downside to corrected decay, like slower convergence early?
>
> We have indeed observed a slight disadvantage of ScionC (constant) when compared to Scion for the simple ViT-S experiments after extensive sweeping. Further experiments with ScionC (cosine) that schedules steady-state norm squared $C^2_{t,l}$ though suggest that the model benefits from later weight norm decrease, not earlier dynamics. We speculate that it might be due to stronger regularization helping generalization. This is now reported in the rebuttal revision.

---

### Official Review · Reviewer_wtvG · 2025-10-29

**Soundness:** 2
**Presentation:** 1
**Contribution:** 1
**Rating:** 2
**Confidence:** 4

**Summary:**

This paper studies the dynamics of weight norms for LLM training and the impact of weight decay hyperparameters. It proposes a weight decay correction for Scion that improves performance for standard language and image model training tasks.

**Strengths:**

The method proposes a weight decay correction for Scion, which seems to improve its performance. Despite its rather confusing presentation, the correction itself comes down to a rather simple correction for lerning-rate scheduling which was proposed before for AdamW by Defazio et al., 2025.

**Weaknesses:**

* It remains unclear what is exactly the goal of this paper, and how the findings are new compared to previous works. To quote from the conclusion:
> A priori there is no reason to believe that it is helpful to induce rapid changes of weight and gradient norms with hyperparameters, intentional or not, and we have not seen evidence to the contrary. [...] weight decay should default to $\gamma^2$.

How is this finding any different/new to the ones from for example Defazio et al., 2025?

* For the proposed correction of Scion, it remains unclear what is exactly the goal of the correction, and why exactly this correction is proposed.
After some simplification of Algorithm 2 for the setting of the experiments (alpha_t is constant), it basically comes down to correcting $\lambda_{t,l} = \lambda_{0,l} * \gamma_{t,l} / \gamma_{max,l}$ for the LR schedule. Altogether, it seems that this is simply the idea of Defazio et al., 2025 applied to Scion. Looking at the experiments in Defazio et al., 2025, this also explains the slightly faster convergence of the corrected method.

* It should be noted that when $\alpha_t$ is constant, then all momentum-related terms in the correction cancel out. No experiments with time-dependent momentum are provided, which seems odd given the strength of claims of the conclusion (*"can be considered a major step in resolving their interactions"*)

* The paper contains many mathematical derivations, where it is unclear if they follow from some assumptions made previosuly, or whether they are true in general. Further, for some arguments, no proofs/derivations are made, and hence it is rather unclear why/when these arguments can be applied due to the lacking mathematical rigor. A list of examples below:

1) Line 045: why is E[u_t^2] constant for large t? Why are $\theta_{t-1}$ and $u_t$ independent (or is this an assumption)?

2) How exactly is steady-state defined? Is this an asymptotic argument for $t\to \infty$? In this case, how does it not conflict with the fact that practical training runs have a finite-horizon LR schedule, and so training always stops after finitely many steps anyway?

3) In Algorithm 1, line 14 the nominator subtracts a scalar $u_{t,l||}$ from a weight matrix/vector. How is this possible? Also, it uses sometimes absolute value notation for a vector, and sometimes norm notation. Please clarify this.

4) line 187: besides the fact that this jumps from Scion to AdamW notation again, how exactly do you arrive at the fact $E[\langle u_{t-1}, u_t\rangle] = 0$?

5) line 189: how can the vector $m_{t,l}$ "decay"? Does this mean its norm decays, and if yes, to what value, and based on which assumptions?

6) Algorithm 2, line 13: if $t$ is the iteration counter, how should one read $t \to \infty$? How can we compute the expectation in the if-clause in practice?

7) line 307: "For the purpose of our experiments, we believe [...]". What do you mean with *believe*? Is this an assumption or an observation from the experiment?


**In summary**, the contributions of this paper seem too incremental to recommend acceptance, as the main idea is to transfer a technique from previous work to a different base optimizer.
For future submissions, I would strongly recommend the authors to revise the motivation and derivations in terms of mathematical rigor.

**Questions:**

* Line 098: "we expect this change to be significant". I did not understand why this renormalization step should have a big impact: if the projection term is small, then it just multiplies with the ratio of norms before and after the usual update of a single step, which should be close to one (in line with the steady-state weight norm assumption). Could the authors clarify what the goals of this experiments are, and what exactly is the conclusion from it and why?

* When reusing the optimal LR from Scion, but extending training length and halving the batch size, this does not longer ensure that the learning-rate for the baseline is chosen optimally. Did you test whether the improvement could be due simply to suboptimal baseline tuning (e.g. by running exactly the same setup as in the Scion paper)?

Minor:

Typo in Alg 2, line 18: should be $\lambda_{t,l}$.

---

> ### Author Response · Authors · 2025-12-04
>
> 1. How is this finding any different/new to the ones from for example Defazio et al., 2025?
>
> Both Defazio (2025) and Kosson et al. (2024) invoke orthogonality and small or zero radial component of the update in their argument for the $\propto \gamma^2$ dependence of the weight norm squared and we show that it's incorrect. In fact, since Defazio (2025) relies on the update rule of AdamW and the properties of the norm layer for its derivation of such dependence and AdamC, one would not expect the result to generalize to other optimizers such as Scion.
>
> 2. No experiments with time-dependent momentum are provided, which seems odd given the strength of claims of the conclusion ("*can be considered a major step in resolving their interactions*")
>
> We did include a set of experiments with time-dependent momentum in the Appendix C, which has been expanded in addition to the  new experiments (Fig. 3) that partially substitute cosine learning rate decay with momentum scheduling in the rebuttal revision.
>
> 3. Why are $\theta_{t-1}$ and $u_t$ independent (or is this an assumption)?
>
> We clearly state that it is an assumption.
>
> 4. Algorithm 2, line 13: if  is the iteration counter, how should one read $t \to \infty$? How can we compute the expectation in the if-clause in practice??
>
> It's just a boolean flag we set. We do not attempt to compute this expectation value during training.
>
> 5. line 307: "For the purpose of our experiments, we believe [...]". What do you mean with believe? Is this an assumption or an observation from the experiment?
>
> It's supported by both experiments ("The Sign norm is stable in both experiments, in support of the hypothesis") and theoretical arguments (Appendix D)
>
> 6. Line 098: "we expect this change to be significant". I did not understand why this renormalization step should have a big impact: if the projection term is small, then it just multiplies with the ratio of norms before and after the usual update of a single step, which should be close to one (in line with the steady-state weight norm assumption). Could the authors clarify what the goals of this experiments are, and what exactly is the conclusion from it and why?
>
> Correct. If the projection is zero, then renormalized AdamW multiplies with the ratio of norms before and after the usual update of a single step. This means that the weight norm will not change at all. The fact that the weight norms of the models trained with renormalized AdamW do change and are nearly the same as the AdamW counterpart shows that the premise is false: The projection i.e. the radial component of the update does not become small.
>
> 7. When reusing the optimal LR from Scion, but extending training length and halving the batch size, this does not longer ensure that the learning-rate for the baseline is chosen optimally. Did you test whether the improvement could be due simply to suboptimal baseline tuning (e.g. by running exactly the same setup as in the Scion paper)?
>
> We acknowledge the lack of retuning for the nanoGPT experiments as a limitation.
>
> We have fixed the Alg 2 typo and now consistently use bold font for vectors to distinguish them from the scalars. We thank reviewer wtvG for pointing them out.

---

### Official Review · Reviewer_hUyc · 2025-11-01

**Soundness:** 1
**Presentation:** 1
**Contribution:** 1
**Rating:** 2
**Confidence:** 3

**Summary:**

In this work, existing insights on decoupled weight decay are challenged. The authors claim that decoupled weight decay proportional to $\gamma^2$ results in a stable weight norm, independent of the optimizer choice. They then apply this insight to propose a corrected weight decay for the Scion optimizer.

**Strengths:**

The authors investigate how decoupled weight decay leads to stable weight (and gradient norms). A better understanding of optimization dynamics and training stability, which this paper aims to address, is a relevant and important topic for improving neural network optimization.

**Weaknesses:**

- The writing is convoluted and hard to follow. The main message of the paper, including the title, is unclear and potentially misleading, leaving it ambiguous whether the main contribution is the proposition of a weight-corrected Scion optimizer or the claimed new insights about weight decay. The main section, however, focuses on the Scion optimizer.
- Claims in the introduction appear to be misleading. In line 096, the authors state: “If the scalar projection $u_{t\parallel}$ is small or zero and the subsequent balanced rotation (Kosson et al., 2024) [...] are important to the training dynamics, we expect this change to be significant.” However, *balanced rotation* in Kosson et al. is a concept of update speed compared across neurons and does not appear to relate to the parallel/perpendicular decomposition the authors use.
- The authors' claim that the perpendicular component's effect on the weight norm is insignificant is poorly validated experimentally. Particularly, they do not explicitly investigate different cases, such as when the weight norm is small and the perpendicular update is large.
- The claim of *improved model performance* is based on insufficient experimental evidence, i.e., they show a single run on ViT-S/16 on ImageNet and a Modded-NanoGPT run on Fineweb-edu-100B. For the latter, they do not include downstream evaluation to show how lower validation perplexity translates to downstream performance.
- Kosson et al. is cited several times as important context, yet the authors do not properly discuss their own work against it in the related work section.

**Questions:**

- Could the authors please clarify the primary contribution of this work? The current framing makes it difficult to assess the main claim.
- Could the authors please address their potential misinterpretation of the *balanced rotation* concept from Kosson et al. (2024), which appears to be a cross-neuron phenomenon, and explain why this work was not discussed in the related work section?
- Regarding Section 1.1: Can the authors provide more evidence that the $u_{t\perp}$ component is *insignificant*? Have they tested other architectures or, as suggested in the review, specific regimes where this component might dominate (e.g., small weight norms and large updates rates)?
- Could the authors provide more evidence for improved model performance, including downstream evaluation of LLM pretraining tasks?

---

> ### Author Response · Authors · 2025-12-04
>
> 1. "Could the authors please clarify the primary contribution of this work? The current framing makes it difficult to assess the main claim."
>
> We experimentally disprove the attribution of $\propto \gamma^2$ dependence of weight norm squared to orthogonality and instead attribute it to independence of the update as a random variable from the weight in the steady state. In the rebuttal revision we further show we can derive the momentum-dependent effective learning rate of the Scion optimizer from the same assumption whose optimal value transfers across different momentum value $\alpha$.
>
> 2. "Could the authors please address their potential misinterpretation of the balanced rotation concept from Kosson et al. (2024), which appears to be a cross-neuron phenomenon, and explain why this work was not discussed in the related work section?"
>
> No, our characterization of Kosson et al. (2024) is not a misinterpretation. While Kosson et al. hedge the language somewhat, it does state that the expected rotation of AdamW is RMS update size divided by equilibrium norm $\hat{\eta_r} = \frac{\hat{\eta_g}}{\widehat{||\mathbf{\omega}||}}$ (Table 1) and that "This approximation is good for small relative updates and a
> relatively small radial component in $\Delta_g \omega$". To the contrary, the renormalized AdamW experiments show that radial component of the update does not become small in the steady state. Also, where does the idea of "cross-neuron phenomenon" come from?
>
> We have already presented the most relevant claims of Kosson et al. (2024) in the previous sections, so we don't find it necessary to mention it again in the related work section.
>
> 3. "Regarding Section 1.1: Can the authors provide more evidence that the $u_{\perp}$ component is insignificant? Have they tested other architectures or, as suggested in the review, specific regimes where this component might dominate (e.g., small weight norms and large updates rates)?"
>
> We mention in our submission ("we cannot exclude the possibility that the balancing effects of AdamW are important for training other classes of models") and only running renormalized AdamW on a class of models as a limitation. We have indeed tried doubling the weight decay coefficient and/or the learning rate in the rebuttal revision and the resulting metrics are still nearly identical to the AdamW counterparts.
>
> 4. "Could the authors provide more evidence for improved model performance, including downstream evaluation of LLM pretraining tasks?"
>
> For the simple ViT-S experiments we have swept the weight decay coefficients and schedule and adjusted our claims accordingly. We acknowledge the lack of downstream evaluation of LLM pretraining tasks as a limitation.

---

### Author Response · Authors · 2025-12-03
**Revision Summary**

Dear reviewers,

We have incorporated your feedbacks and revised the manuscript. The revisions of the main text are along 3 major axes and highlighted in blue:

1. Implications

In order to explore the parameter space of different momentum values and non-constant momentum $\alpha_t$, we derived and empirically verified that the Total Update Contribution (TUC) of a minibatch under the Scion optimizer is better characterized by the momentum-dependent effective learning rate $\gamma_\mathrm{eff} \coloneq \gamma \sqrt{\frac{2 - \alpha}{\alpha}}$ instead of the raw learning rate $\gamma$. Not only does the optimal $\gamma_\mathrm{eff}$ transfers across different momentum values, but we can also mimic the effects of cosine learning rate decay by making $\gamma_\mathrm{eff}$ follow the cosine decay by increasing the value of $\alpha_t$ while the raw learning rate $\gamma$ stay constant during most of the experiment.

2. Rigor

Besides that Renormalized AdamW experiments now consist of 3 pairs of comparisons, we have swept the weight decay coefficients for the simple ViT-S ImageNet-1k experiments for the Scion and ScionC experiments. To our surprise the ScionC baseline holds a slight edge over ScionC (constant) that is designed to keep weight norms constant in the steady state. We have further explored the experiments that schedule the expected weight norms and verified that induced weight norm decreases turn out to be beneficial in this case. We have therefore mostly restricted our claims to understanding the training dynamics of optimizers with decoupled weight decay that allow us to exert better control.

3. Notation / Formulation

We have fixed some typos and followed the suggestion of reviwer M9jm to use bold symbols for vectors to distinguish them from scalars. In addition, we have revised the formulation of ScionC to specify the steady-state norm squared $C^2_{l}$ directly and include the option of scheduling $C^2_{t,l}$. Under constant momentum $\alpha$ the old formulation can be parameterized to be equivalent, but not with momentum scheduling $\alpha_t$. The new formulation is also simpler and arguably more intuitive.

We will answer your questions individually, though due to the circumstances it's only for the record.

---

### Meta-Review · Area_Chair_wsfC · 2026-01-05

**Summary:**

The reviewers broadly agree that the paper addresses a relevant and timely topic, understanding the role of decoupled weight decay and its interaction with optimization dynamics, but raises substantial concerns that ultimately inform a negative recommendation. Across reviews, there is a shared view that the paper’s main contribution is unclear and largely incremental relative to prior work, particularly Defazio et al. (2025) and Kosson et al. (2024). Although the authors propose a different explanatory framework based on a steady-state independence assumption, the practical conclusion (weight decay scaling proportional to the square of the learning rate) is the same as in prior work, and reviewers are not convinced that revising the interpretation alone constitutes a significant advance. In addition, reviewers consistently question the strength and applicability of the theoretical assumptions and find the experimental evidence insufficient in scope and depth to support the paper’s claims.

**Reviewer Concerns:**

Several reviewer concerns were partially addressed in the rebuttal. The authors clarified that the independence between weights and updates is an explicit assumption, corrected notational inconsistencies and typos, and provided additional explanations and experiments related to momentum scheduling. The rebuttal also acknowledges multiple limitations raised by reviewers, including limited experimental scale, lack of downstream evaluation for language models, and incomplete baseline retuning.

However, the core concerns shared by reviewers largely remain outstanding. First, the question of novelty is unresolved: reviewers remain unconvinced that the work offers a clear, substantive contribution beyond existing results, as the main practical prescription coincides with prior work and the claimed novelty lies primarily in an alternative explanatory assumption. Second, the steady-state independence assumption is still viewed as strong and insufficiently justified, particularly in the presence of momentum, learning-rate schedules, and finite training horizons common in practice. The additional empirical evidence provided does not fully address doubts about when and why this assumption should hold. Third, the experimental validation remains limited, relying on relatively small models and a narrow set of settings, with potential confounding effects from hyperparameter choices and without robust ablations or downstream evaluations. Finally, despite improvements in clarity, reviewers continue to find the overall framing and positioning with respect to prior work unconvincing and, at times, overstated.

**Reviewer Scores:**

Given the rebuttal and discussion, it is unlikely that any reviewer would substantially change their original evaluation. Reviewers who raised concerns about unclear contribution, theoretical assumptions, and limited experiments would likely maintain their rejection recommendations. Even the relatively more positive review remains constrained by unresolved issues around assumption validity and experimental scope, and would likely not shift to a positive score.

---

### Decision · Program_Chairs · 2026-01-26

Reject